# GINN: Fast G̲PU-TEE Based I̲ntegrity for N̲eural N̲etwork Training

## ABSTRACT

Machine learning models based on Deep Neural Networks (DNNs) are increasingly deployed in a wide range of applications, ranging from self-driving cars to COVID-19 treatment discovery. To support the computational power necessary to learn a DNN, cloud environments with dedicated hardware support have emerged as critical infrastructure. However, there are many integrity challenges associated with outsourcing computation. Various approaches have been developed to address these challenges, building on trusted execution environments (TEE). Yet, no existing approach scales up to support realistic integrity-preserving DNN model training for heavy workloads (deep architectures and millions of training examples) without sustaining a significant performance hit. To mitigate the time gap between pure TEE (full integrity) and pure GPU (no integrity), we combine random verification of selected computation steps with systematic adjustments of DNN hyperparameters (e.g., a narrow gradient clipping range), hence limiting the attacker's ability to shift the model parameters significantly provided that the step is not selected for verification during its training phase. Experimental results show the new approach achieves 2X to 20X performance improvement over pure TEE based solution while guaranteeing a very high probability of integrity (e.g., 0.999) with respect to state-of-the-art DNN backdoor attacks.

## 1 INTRODUCTION

Every day, Deep Learning (DL) is incorporated into some new aspects of the society. As a result, numerous industries increasingly rely on DL models to make decisions, ranging from computer vision to natural language processing. The training process for these DL models requires a substantial quantity of computational resources (often in a distributed fashion) for training, which traditional CPUs are unable to fulfill. Hence, special hardware, with massive parallel computing capabilities such as GPUs, is often utilized Shi et al. (2016). At the same time, the DL model building process is increasingly outsourced to the cloud. This is natural, as applying cloud services (e.g., Amazon EC2, Microsoft Azure or Google Cloud) for DL training can be more fiscally palatable for companies by enabling them to focus on the software aspect of their products. Nevertheless, such outsourcing raises numerous concerns with respect to the privacy and integrity of the learned models. In recognition of the privacy and integrity concerns around DL (and Machine Learning (ML) in general), a considerable amount of research has been dedicated to applied cryptography, in three general areas: 1) *Multi-Party Computation (MPC)* (e.g., Mohassel & Zhang (2017)), 2) *Homomorphic Encryption (HE)* (e.g., Gilad-Bachrach et al. (2016)), and 3) *Trusted Execution Environment (TEE)* (e.g., Hunt et al. (2018); Hynes et al. (2018)). However, the majority of these investigations are limited in that: 1) they are only applicable to simple shallow network models, 2) they are evaluated with datasets that have a small number of records (such as MNIST LeCun & Cortes (2010) and CIFAR10 Krizhevsky et al.), and 3) they incur a substantial amount of overhead that is unacceptable for real-life DL training workloads. In their effort to mitigate some of these problems, and securely move from CPUs to GPUs, *Slalom* Tramèr & Boneh (2019) mainly focus on the computational integrity at the *test* phase while depending on the application context. It can also support enhanced data privacy, however, at a much greater performance cost.

To address these limitations, we introduce **GINN** (See Figure 1); a framework for *integrity-preserving* learning as a service that provides integrity guarantees in outsourced DL model training in TEEs. We assume that only the TEE running in the cloud is trusted, and all the other resources

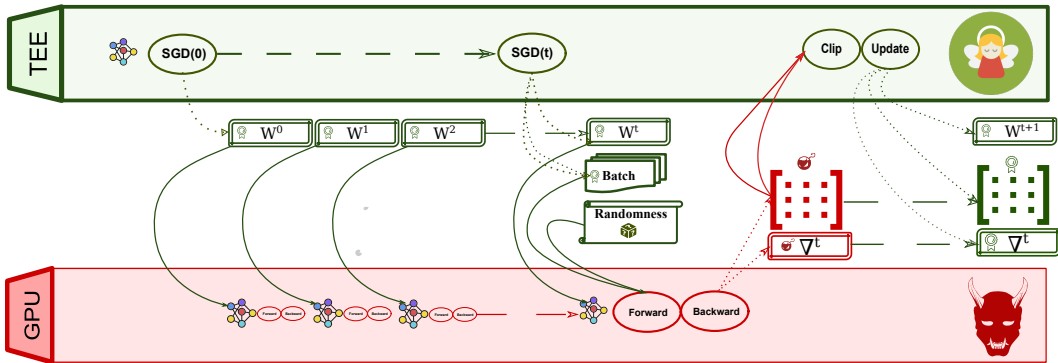

Figure 1: The main architecture of **GINN**. The TEE handles mini-batch selection, layer-specific randomness, and parameter initialization. The GPU performs forward and backward passes over the mini-batch (items selected by SGX provided seed) and reports the computed gradients to the TEE. TEE then clips the gradients and performs the weight update. Also, TEE preserves the MAC-authenticated intermediate gradient reports. During verification, TEE performs the forward and backward passes with the batch items along with layer-specific randomness (regenerated) and compares the gradients with the GPU's report.

such as GPUs can be controlled by an attacker to launch an attack (e.g., insert a trojan). In this context, our goal is to support the realistic deep learning training workloads while ensuring data and model integrity. To achieve this goal, we focus on the settings where maintaining the learning process's integrity is critical, while the training data may not contain privacy sensitive information. For example, we may want to build a traffic sign detection model on public traffic sign images and may still like to prevent attacks that can insert trojan during the training phase. Furthermore, we want to provide assurances that the model is trained on the specified dataset, with known parameters so that the performance of the model can be replicated and audited for accountability and integrity.

The trivial approach of executing the entire learning process inside a TEE is not scalable since TEEs are much slower compared to GPUs. Furthermore, even the existing performance improvement techniques (e.g., random matrix verification provided in Tramèr & Boneh (2019)) are not enough to scale up to large DL model learning settings.

To alleviate the TEE bottleneck, we propose incorporating *random verification* of the computation steps. This strategy is based on the observation that it is unnecessary to verify all of the GPU's computation steps. Rather, we only need to verify occasionally to catch any deviation with a very high likelihood. Given that random verification may itself be insufficient (theoretically, an attacker can launch a successful attack by modifying only a single unconstrained gradient update), we further show how parts of the DL hyperparameter setting process, such as *clipping rate* should be modified to prevent single step attacks, and require a larger number of malicious updates by an attacker that controls the GPU. Simply, **GINN** limits the amount of change an adversary can inflict on a model through a single SGD step. As a consequence, the adversary is forced to keep attacking while randomly being verified by the TEE. Using the state-of-the-art backdoor attacks, we illustrate that random verification technique can detect attacks with a high probability (e.g., 0.99) while enabling 2x-20x performance gains compared to pure TEE based solutions.

**The specific contributions of this paper are as follows:**

- We introduce the first approach to support integrity-preserving DL training by random verification of stochastic gradient (SGD) steps inside TEE to ensure the integrity of training pipeline data, parameters, computation function, etc. with a high probability.

- We illustrate how gradient clipping can be used as a defensive measure against single (or infrequent) step attack in combination with random verification.

- We show the effectiveness of our TEE random verification and gradient clipping through extensive experimentation on DNN backdoor attacks.

## 2    BACKGROUND AND RELATED WORKS

Our system combines deep learning training on specialized fast hardware such as Graphical Processing Units (GPU) with Intel Software Guard Extensions (SGX) based TEE to ensure the produced model's integrity. Details on SGD training and gradient clipping are provided in Appendices B and C.

### 2.1    ATTACKS ON DNN MODELS IN TRAINING PHASE

Attacks on DNN models can be realized during both *training* or *test* phases. However, GINN is concerned with integrity/accountability issues during the training phase of DNN models, such that attacks related to testing are out of the scope of this paper since test time attacks have been addressed before (e.g., *Slalom* Tramèr & Boneh (2019)). In the literature, particularly in the computer vision domain, targeted trojan attacks on DNN classification models have become a real concern as deep learning has grown in its adoption. These attacks tend to alter the prediction of models if a specific condition in the input is met. These conditions may be *feature-based* Gu et al. (2017); Liu et al. (2017); Turner et al. (2018) or *instance-based* Chen et al. (2017); Shafahi et al. (2018). Recently, trojan attacks have been extended to Reinforcement Learning (RL) and text classification models Panagiota Kiourti (2019); Sun (2020).

In practice, these attacks are implemented by manipulating samples during training through data poisoning. For instance, stamping images with a pattern and modifying its label. Interestingly, these models provide similar competitive classification test accuracy compared to clean models (i.e., models have not been attacked). As a consequence, it is non-trivial to distinguish these trojaned models from non-trojaned ones based on model accuracy alone. To make matters worse, even if the model owner was aware of examples of the trojan trigger pattern, the owner would need to patch the model through re-training to dampen the efficacy of the trojan trigger pattern. Retraining does not always guarantee complete removal of the trojan behavior from the model. To date, various techniques have been proposed to diagnose and mitigate of trojaned models. However, all approaches are either based on unrealistic assumptions or are excessively costly. For instance, the Neural Cleanse Wang et al. (2019) requires access to a sizable sample of clean inputs to reverse-engineer the backdoor and has shown to be successful only for trigger patterns with a relatively small size. ABS Liu et al. (2019) improves upon Neural Cleanse in that requires a significantly smaller number of samples; however, it assumes that the responsible trojan neurons can activate trojan behavior independently from each other, which is unlikely to be true in practice.

Attacking the training pipeline to inject a trojan(s) in the final model is the cheapest and, thus, is likely the most desirable form of attack for real-world adversaries to launch. As such, throughout this work, we mainly focus on showing our methods' effectiveness in preventing this type of attack from happening. It should be noted that our method is *orthogonal* to *attack* type and is sufficiently *generic* to catch any continuous attack during the training of a DNN model. GINN relies upon *proactive* training as opposed to post-training or deployment-time methods to assess the health of a DNN model. As we explain later in section 3, we assume that the initial training dataset is provided by an honest user and is free of manipulation. With this as a basis, GINN limits the amount of change an adversary can inflict on a model through a single SGD step. As a consequence, the adversary is forced to keep attacking while randomly being verified by the TEE.

### 2.2    INTEGRITY FOR DNN TRAINING

GINN's main goal is to enable high-integrity training pipeline so that end users are assured that the model is built on the specified dataset, using specified parameters without modification. Thus, the final model users know who built the model, what dataset was used for training, and what algorithms were put in place for building the model. If, at any point during training, GINN GINN detects a deviation from the specified execution, it will not sign the final model to ascertain its validity. Tramèr & Boneh (2019) took a first step towards achieving both *fast* and *reliable* execution in the *test* phase but neglected the training phase. The training phase is far more computationally demanding than the test phase, such that verification of all steps in training requires a substantially longer time. Since parameters keep changing, we cannot benefit from pre-computation. Second, backward pass involves computing gradients for both the inputs and the parameters and takes longer than forward

pass. Despite the mentioned hurdles, as our investigation shows, it may not be necessary to verify every step to achieve integrity guarantees with high probability.

## 2.3 INTEL SGX

SGX Costan & Devadas (2016) is an example of a common TEE that is available in many modern-day computers. As outlined in Table 2, it provides a secluded hardware reserved area, namely, processor reserved memory (PRM), that is kept private (i.e., it is not readable in plaintext) from the host, or any privileged processes, and is free from *direct* undetected tampering. It also supports *remote attestation*, such that users can attest the platform and the running code within enclave before provisioning their secrets to a remote server. Calls from routines that should transition to/from enclave are handled through predefined entry points that are called Ecall/Ocall that must be defined in advance, before building the enclave image. While it provides security and privacy for numerous applications (e.g., Priebe et al. (2018); Shaon et al. (2017); Kunkel et al. (2019)), due to its limited memory and computational capacity, directly running unmodified applications inside SGX can induce a significant hit on performance. This is especially the case for applications that require large amounts of memory, such as training DNNs.

## 3 THREAT MODEL

Attacks on the integrity of DNNs can be orchestrated at different stages of the model learning pipeline (e.g., data collection or training). We assume the TEE node in GINN is trusted, and the bytes stored on the *PRM* are always encrypted and authenticated before they are fetched inside the CPU. We assume that the data sent to GINN comes from honest users via a secure/authenticated channel and is devoid of malicious samples.[1] For the training phase, we assume that the adversary has complete knowledge about the network structure, learning algorithm, and inputs (after TEE performs an initial pre-processing) to the model. In our threat model, the adversary is in complete control of the host system's software stack, and hardware (unprotected RAM, GPU), except for the CPU package and its internals. Therefore, the code that runs inside the enclave is free from tampering, and the data that is accessed inside the cache-lines or registers are not accessible to the adversary. For the inputs supplied to DNN tasks, the adversary is capable of performing insertion, modification, and deletion to influence the final model towards her advantage. As a result, an attacker may report wrong gradients as opposed to correctly computed ones.

## 4 SYSTEM DESIGN

GINN offers integrity and accountability for the training phase of a DNN model while inducing limited computational overhead. An overview of GINN is illustrated in figure 1, and we refer the reader to table 2 for symbol descriptions and abbreviations.

Before the training phase initiates, the training dataset is decrypted and validated inside the TEE. Besides, for each SGD step, the randomness regarding the mini-batch selection and the network layers Srivastava et al. (2014) are derived within the TEE and supplied to the GPU. As a result, the adversary will face a more constrained environment. Moreover, in our design, the GPU always performs a forward and a backward pass and reports the computed gradients to the TEE. At this point, the TEE clips the gradients and updates the parameters (low overhead operation). Finally, GINN randomly decides to verify the SGD steps within the TEE or not. GINN is optimized to guarantee a high level of integrity and the correctness of the model while providing an infrastructure that does not suffer from the substantial computational requirements of pure TEE-based solutions. We assume an honest and authenticated user will send her data encryption key $K_{client}$ (after remote-attestation) to the TEE. Next, the TEE decrypts/verifies the initial encrypted dataset using the $K_{client}$ and supplies the trainer (GPU) the plain-text of the training set. If the TEE fails to detect any violations of the protocol during training, it will sign a message that certifies the final model.(Please see Appendix A for more details of the signed message). Then, during testing and deployment, the user can verify the digital signature of the model.

---

[1]Detecting malicious samples is beyond the scope of this work.

**Training with GINN** At the beginning of mini-batch iteration **i**, TEE supplies the untrusted GPU with the randomness for that iteration. After completion of the forward and backward passes over the mini-batch, the computed gradients are sent to the TEE for clipping and updating the parameters. Next, the TEE integrates the clipped gradients with the parameters of the previous step. GINN always clips the reported gradients and ensures that they are within a narrow range *so that evolving the model towards the attacker's intended model requires a prolonged malicious intervention by the attacker*. GINN accepts the reported gradients and only applies the clipped version to the snapshot taken at the specific iteration. If the computation at that step is selected for random verification, then the faulty behavior can be detected. If not, the chance that the model evolved towards the attacker's desired optima will likely require multiple rounds providing ample opportunity for detection. The verification is done randomly to prevent an attacker from guessing which step is verified.

**Probabilistic Verification with GINN** The TEE randomly decides whether or not to verify the computation over each mini-batch. If the mini-batch is selected for verification, then the intermediate results are saved, and the verification task is pushed into a verification queue. *Verification by the TEE can take place asynchronously* and it does not halt the computation for future iterations on the GPU. The authenticity of snapshots is always verified with a key that is derived from a combination of the TEE's session key, $SK_{SGX}^{session}$ and the corresponding iteration. When the TEE verifies step **i**, it populates the network parameters with the snapshot it created for the step **i − 1**. It then regenerates the randomness from step $i$ to obtain the batch indices and correctly sets up the per layer randomness. Given that the TEE's goal is to verify that the reported gradients for step $i$ are correctly computed, GINN does not keep track of the activation results. Rather it only requires the computed gradients, batch mean/std (for BatchNorm layer), and matrix multiplication outcomes (in case random matrix multiplication verification is chosen). These required parameters are saved for verification.

**Randomized Matrix Multiplication Verification with GINN** Matrix Multiplications (MM) take up the bulk of the resource-heavy computations in DNNs. In modern DNN frameworks, convolutional and connected layers computation is implemented in the form of a matrix multiplication in both of the forward and backward passes. Table 4 in Appendix G depicts the computations in the forward pass and backward gradient with respect to the weights and previous layers' outputs in the form of a rank 2 tensor multiplication. Fortunately, there exists an efficient verification algorithm for matrix multiplication( Freivalds (1977)) when the elements of matrices belong to a field. In this work, we leverage these random matrix multiplication verification algorithms as well.

## 5 INTEGRITY ANALYSIS

To achieve the integrity goal, we need to derive the probability $p_v$ (i.e., the verification probability of each step) to achieve our integrity goal $p_i$ (i.e., the probability that attacker can modify the result without being detected is less than $1 - p_i$).

**Definition 1.** Assuming the DNN training requires a total of $B$ steps, for each step report ($R_b \ \forall b \in [1, B] \ \land R_b \in \{0, 1\}$) has a probability $p_c$ for being corrupted ( i.e., $R_b = 1$), and the overall integrity requirement probability goal $p_i$ (for example $p_i = 0.999$).

**Theorem 1.** *Given a total of B steps during SGD training, the required probability of choosing a step to verify ($p_v$) should be greater than* $B^{-1}(\frac{\log(1-p_i)}{\log(1-p_c)} - 1)$.

*Remark.* Given each step contains $m$ independent matrix multiplication (MM) operations that is to be repeated $k$ times (independently), and each random entry is chosen from a field of size $|S|$, the probability of error (accepting a wrong MM equality) is less than $\alpha = \frac{1}{|S|^{mk}}$ Freivalds (1977).

**Theorem 2.** *If random matrix multiplication verification is used, given the configuration of Theorem 1, the required probability of choosing a step to verify ($p_v$) should be greater than* $B^{-1}(\frac{\log(1-p_i)}{\log((1+(\alpha-1)p_c)} - 1)$.

We refer the reader to Appendix D.1 and D.2 for complete proofs of the theorems. The threshold probability in Theorem 2 yields approximately the same values as Theorem 1 when $\alpha \to 0$. However, the randomized matrix multiplication verification requires a $O(N^2)$ operations (assuming two $N \times N$ matrices) compared to regular matrix multiplication that requires $O(N^3)$ operations.

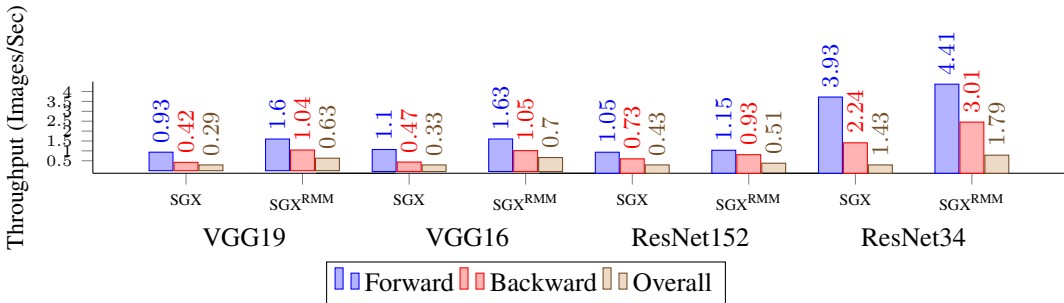

Figure 2: Throughput of the SGD training step for VGG19,VGG16, ResNet152, and Resnet34 on ImageNet dataset with repect to forward and backward passes. RMM can lead to verification that is twice as fast as full MM verification in case of a VGG architecture.

Table 1: Trained models with restricted clipping and learning rate

| Dataset | clip | lr | %clean | %attack | total |
|---------|------|-----|--------|---------|-------|
| GTSRB | $10^{-4}$ | $10^{-4}$ | $\geq \%95$ | $\geq \%85$ | 124 |
| MNIST | $10^{-4}$ | $5 \times 10^{-5}$ | $\geq \%95$ | $\geq \%85$ | 49 |
| CIFAR10 | $5 \times 10^{-4}$ | $10^{-1}$ | $\geq \%90$ | $\geq \%85$ | 94 |

## 6    EXPERIMENTAL EVALUATION

We executed our experiments on a server with Linux OS, Intel Xeon CPU E3-1275 v6@3.80GHz, 64GB of RAM and an NVIDIA Quadro P5000 GPU with 16GB of memory. Our attack code is implemented in python 3.6 using the pytorch Paszke et al. (2019) library. We use Intel SGX as our platform for TEE. For SGX proof-of-concept implementation, DarkNet Redmon (2013–2016) library is significantly modified to run the experiments. Our SGX code has been tested with SDK 2.9 and the code runs inside a docker container in hardware mode. Our experiments are designed to investigate two aspects. First, we evaluate the impact of integrating randomized matrix verification and show that it can potentially increase computational efficiency.Second, we analyze the effectiveness of gradient clipping in forcing the attacker to deviate from the honest protocol in significant numbers of mini-batch steps. Various attack hyper-parameters (e.g. poisoning rate) were evaluated in determining their importance towards a successful attack with minimal deviation. This is important because, if the attacker needs to deviate over more mini-batch steps, then the TEE can detect such deviations with a smaller number of random verification steps (i.e., $p_c$ is higher in equation 2 ).

**TEE Performance**    We tested TEE performance on VGG16, VGG19 (Simonyan & Zisserman (2014)), ResNet152, and ResNet34 (He et al. (2016)) architectures with the ImageNet (Deng et al. (2009)) dataset (see Appendix F.2 for results with CIFAR10 dataset( Krizhevsky et al.)). Figure 2 illustrates the throughput of deep networks. Usually, most of the computation takes place in the convolution layers of the network. However, the backward pass on average yields a smaller throughput since it involves one more MM (weight gradients, and input gradients) than the forward pass which only invokes MM once (i.e., output of convolution or fully-connected layers). The implementation is quite efficient in terms of MM operations which uses both vectorized instructions along with multithreading. Both VGG networks gives better improvements ($\approx 2.2\mathbf{X}$) than ResNet ($\approx 1.2\mathbf{X}$). Considering the TEE randomly decides to verify a SGD step with probability $p_v$, the overall improvement is approximately multiplied by $\frac{1}{p_v}\mathbf{X}$. For instance, if we assume the attacker only requires to deviate with probability $5.10^{-5}$ (i.e., deviating in five steps out of a hundred thousand of steps), we can detect it with $p_v \approx 0.1$ (See Figure 5d in Appendix). For VGG networks it means approximately 22$\mathbf{X}$ improvements compared to a pure TEE-based solution that verifies every step. Nonetheless, as our experiments suggest, we believe that attacking these deep models that are trained on massive datasets should require a significantly higher number of deviations. This in return may result in a much higher performance gain.

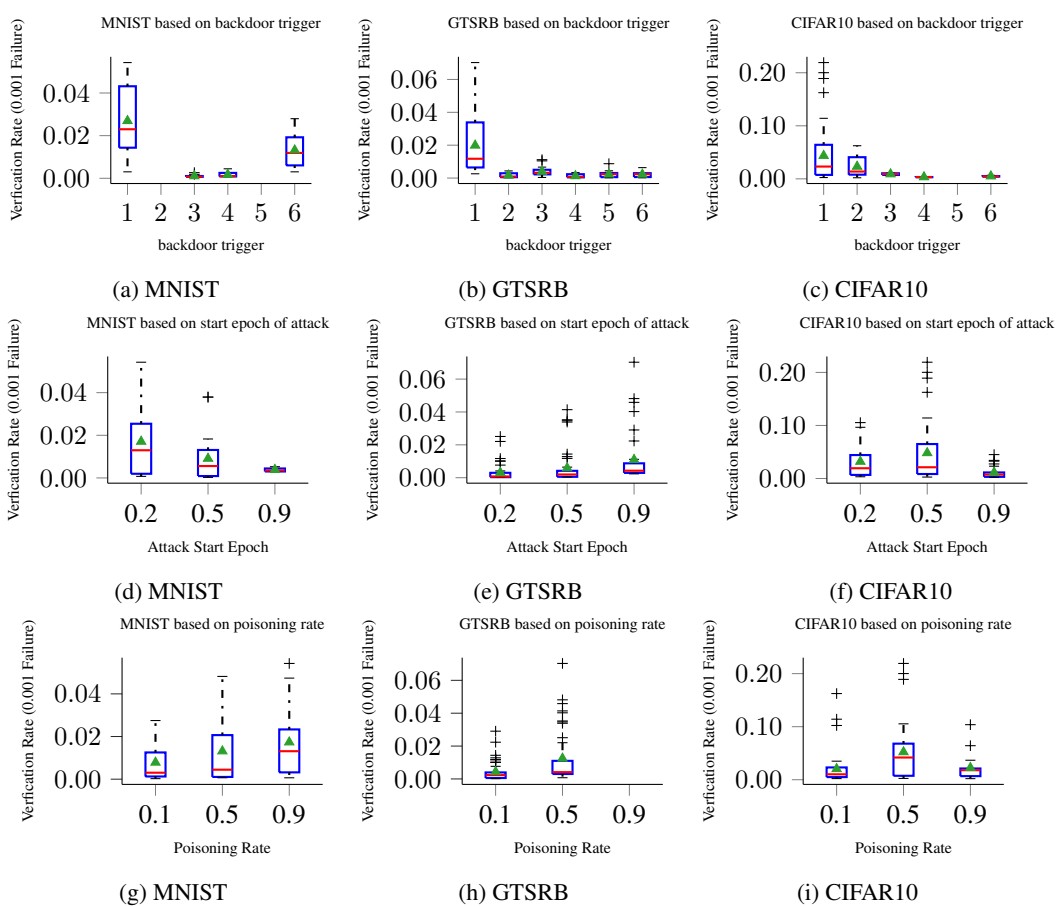

Figure 3: A boxplot representation of the verification rates required by TEE for detection failure $\leq 10^{-3}$. Based on the attack hyper-parameters: 1) backdoor trigger type (first row). 2) epoch that attack starts (second row, $20\%\times, 50\%\times$, and $90\%\times|epochs|$). 3) backdoor poisoning ratio

**Combined Impact of Gradient Clipping and Learning Rate on Attack Success** As shown in Table 1, we selected models that achieved high performance on both clean and poisoned test samples. We conducted the attack in the following manner. [2] First, the trainer follows the correct protocol (e.g. learning with mini-batch SGD) until $epoch_{attack}$. At this epoch, the attacker starts by injecting a certain number of poisoned samples ($pois_{rate} \times batch\_size$) from every class into the training batch and labels it as the target label. The attacker continues to attack until they achieve a desired threshold in terms of success rate (correctly classifying backdoored samples as the attacker's target label). Once it passes the threshold, the attacker halts the attack and returns to the honest protocol, while observing the decay in attack success rate. If the success rate falls below the desired low-threshold, the attacker transitions back to attack mode and repeats the aforementioned strategy. CIFAR10 (Krizhevsky et al.), GTSRB (Stallkamp et al. (2012)) and MNIST (LeCun & Cortes (2010)), were used to analyze the impact of multiple factors imposed by the attacker that also aims to maintain an acceptable accuracy over clean test set. For MNIST, and GTSRB the *Adam* Kingma & Ba (2014) optimizer (which requires a smaller initial learning rate), and for the CIFAR10 dataset, *SGD* with momentum are used. Additionally, for MNIST, and GTSRB $10\%$ of the training set was chosen for the validation set to help adjust the learning rate based on the validation set loss. Save for CIFAR10 dataset, the learning rate was set to decay (by tenfold) at fixed epochs $(40, 70, 100)$.

**Backdoor Trigger Pattern** We applied 6 different backdoor triggers ( all of them are shown in Appendix, Figure 9). The MNIST dataset only has single channel images, we converted it to a

---

[2]We also experimented with the scenario where an attacker attack at steps chosen randomly. This type of attack was not successful. Hence we do not report those results here.

three channel image to apply the triggers 3 to 6. As shown in Figures 3 (a–c) the trigger pattern can significantly influence the effectiveness of the attack. The red lines show the median, while the green dots correspond to the mean. In all of the datasets, the first trigger pattern (Figure 9a) was the most effective one. It covers a wider range of pixels compared to the second trigger type (Figure 9b). As a consequence, it is more likely for the model to remember the trigger pattern across longer periods of SGD steps. Moreover, because photo filters (e.g. *Instagram*) are popular these days, we investigate the possibility of conducting attacks using some of the filters(or transformations) as the trigger pattern. However, covering a very wide range of pixels does not lead to a stealthy attack, as illustrated for the last four patterns. These patterns cover the whole input space and transform it to a new one that they share a lot of spacial similarities while only different in tone or scale (e.g. Figure 9d). Learning to distinguish inputs that are similar, but only different in their tone is demanding in terms of continuity of the attack. In this case, both of the images (w. and w.o. the trigger) are influencing most of the parameters and filters in a contradictory manner (different classification label). Thus it takes a significant number of steps for the network to learn to distinguish them when gradient clipping is applied.

**Attack Start Epoch**    Another major factor influencing the evasiveness of the attack is when attacker starts the attack. For instance, early in the training phase the learning rate will be high, such that a savvy attacker might believe they can avoid low clipping values by initiating their attack. However, if the attack begins too early, then it is unlikely that the model has yet converged. Therefore, beginning the attack too early may require an unnecessarily high number of (unnecessarily) poisoned batches, which, in turn, would raise the probability of detection. Yet even if the attacker was successful, once they halt the attack, the model will likely evolve the parameters back to a clean state relatively quickly, and may require the attacker to re-initiate their attack. At the same time, given that the system uses a low clipping value, if the attacker waits toward the end of training, the attack is again unlikely to be effective. It would be unlikely that the attack succeeds before the end of the training; particularly due to having a considerably smaller learning rate. As shown in Figures 3 (d–f), the best time for the attack is when the model has a relatively low loss on clean training inputs, and the combination of learning rate and clipping value (effective attainable update) could yield the model to move toward attacker's desired optima. However, for MNIST, which is an easier learning task, attacking early gives the attacker a better chance to launch a stealthier attack. We speculate that this is due to the fast convergence of the model. After a few epochs, it quickly reaches a stable low training loss value for clean images. As a result, when the attacker concludes the attack (after reaching to the desired threshold), it is generally preserved far longer than the other two datasets.

**Mini-Batch Poisoning Ratio**    As we stated earlier, the $pois_{rate}$ parameter is the ratio of the number of poisoned samples in the batch to the batch size. It is one of the critical factors amongst those we investigated. Especially when gradient clipping is used, setting $pois_{rate}$ appropriately can help the attacker by moving more parameters toward the desired optima. However going beyond the ratio 0.9 (i.e., $pois_{rate} > 0.9$ can impact the training negatively for both clean inputs and poisoned inputs. As depicted in Figures 3 (g–i), our experiments suggest that filling more than half the batch with poisoned samples seems to be effective across all datasets. Although for the MNIST dataset, it shows that higher values can slightly perform better, but this is not confirmed by experiments on more complex datasets. For example, for the GTSRB dataset, we could not find a successful attack model with acceptable clean input accuracy.

**Further Analysis On ImageNet Dataset**    Due to the computational resource limit, we only investigate a few instances of our attack on ResNet34 architecture over the ImageNet dataset. The state of the art reference (pytorch Paszke et al. (2019) repository) achieves an accuracy of 0.73. However, we only reinitialize and train the fully-connected plus the last four convolutional layers (10M out of 21M parameters) with batches of size 256. The initial learning rate is set to 0.05, and $pois_{rate}$ is 0.5. Moreover, the first trigger pattern (Figure 9a) is used as the only pattern (resized to 256x256). Finally, our attack starts at the beginning of the third epoch.

**Clean Training For 100K Steps**    In order to see the optimal accuracy that can be achieved for current setting, we selected three varying clipping values, i.e., unbounded, high (2.0), and low (0.0005). Training with unbounded clipping value was futile and the model barely reached 0.01

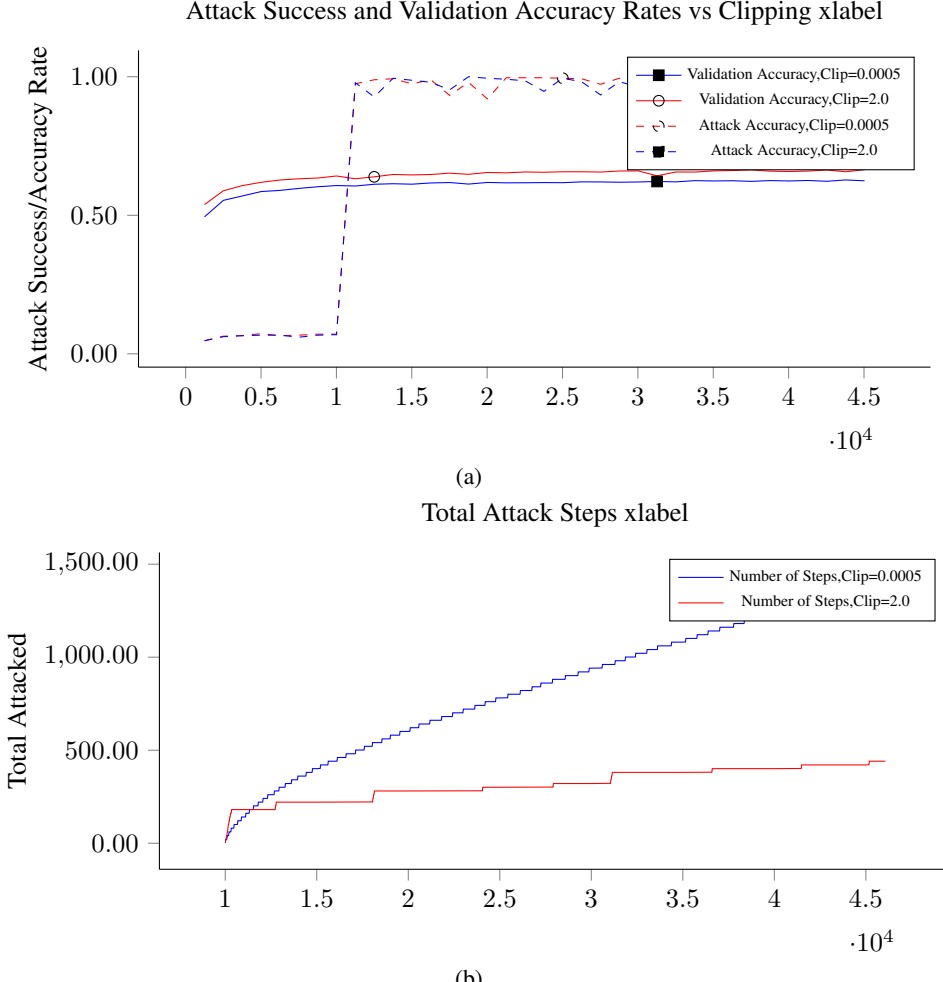

(a)

(b)

accuracy. However, training with the high clipping value converged to an acceptable optima of 0.68 accuracy. Please note that we only trained the model for far less iterations than what is usual for ImageNet (1M-2M steps) and the learning rate was fixed. Additionally, for the low clipping value, the model converged to an optima of 0.64 accuracy. We want to point out that having a learning decay schedule will make the accuracy approach the 0.73 (similar to the reference). However, our goal was to analyze the impact of clipping on the attack for a large dataset.

**Gradient Clipping Impact on Poisoning Backdoor Attack Evasiveness** We conducted the attack for a subset of source labels (10 out of 1000 classes) to a single target class. This attack requires less interference from the adversary because she does not need to change all classes to recognize the backdoor trigger, influencing a smaller subset of parameters. Figure. **??** shows how smaller clipping rate forces the attacker to continue the attack extensively, which will result in a small verification rate.

## 7 CONCLUSION

This paper introduced the GINN system, which provides integrity in outsourced DNN training using TEEs. As our experimental investigation illustrates, GINN scales up to realistic workloads by randomizing both mini-batch verification and matrix multiplication to achieve integrity guarantees with a high probability. We have further shown that random verification in combination with hyperparameter adjustment (e.g., setting low clipping rates), can achieve 2**X**-20**X** performance improvements in

comparison to pure TEE-based solutions while catching potential integrity violations with a very a high probability.

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

## A    MODEL SIGNING BY TEE

We assume, an honest and authenticated user will send her data encryption key $K_{client}$ (after remote-attestation) to the TEE. Next, the TEE decrypts and verifies the initial encrypted dataset using the $K_{client}$ and supplies the trainer (GPU) the plain-text of the training set. If the TEE fails to detect any violations of the protocol during training, it will sign the following message that certifies the final model where $\mathcal{W}$ is the model parameters, $ds$ is training dataset, $v\_num$ is the model version number, $SHA256$ is Sha 256 bit cryptographic hash function,

$$SHA256(SHA256(\mathcal{W})||v\_num||SHA256(ds)||Sig_{client}^v)$$

with signature key $Sig_{SGX}^s$ of the enclave

Table 2: Symbols and Acronyms Description

| Category | Symbol | Description |
|---|---|---|
| TEE | $K_{SGX}^{session}$ | TEE's session key for learning task |
| | $Sig_{SGX}^s$ | SGX signature signing key |
| | $K_{client}$ | client's encryption key |
| | $Sig_{client}^v$ | clients public key |
| | PRM | Processor Reserved Memory |
| | EPC | Enclave Page Cache |
| Neural Network | RMM | Randomized Matrix Multiplication |
| | FV | Full Verification (No RMM) |
| | DNN | Deep Neural Network |
| General | $\mathcal{W}$ | model parameters |
| | $ds$ | training dataset |
| | $v\_num$ | model version number |

## B    DEEP LEARNING TRAINING

In the recent decade, Deep Neural Networks (DNN) gained an enormous attraction in solving problems related to computer vision and natural language processing Krizhevsky et al. (2012); Simonyan & Zisserman (2014); He et al. (2016); Szegedy et al. (2015; 2017). In practice, these networks are stacks of layers that each perform a transformation $\mathcal{F}_{\mathcal{W}}^l(\cdot) \ \forall l \in |L|$ where $\mathcal{X}^{l+1} = \mathcal{F}_{\mathcal{W}}^l(\mathcal{X}^l)$ and $|L|$ is the number of layers. The training task is to learn the correct parameters (point-estimates) $\mathcal{W}^*$ that optimizes (commonly minimizes) a task-specific (e.g. classification) loss function $\mathcal{L}$.

The most common way of training the DNN's parameters is through mini-batch Stochastic Gradient Descent (SGD) Robbins & Monro (1951). A randomly selected mini-batch of dataset is fed to the DNN and the value of objective function $\mathcal{L}$ is calculated. This is usually called the *forward* pass. Next, to derive the partial gradients of $\mathcal{L}$ w.r.t $\mathcal{W}$ ($\nabla_{\mathcal{W}}^{\mathcal{L}}$), a *backward* pass is performed Goodfellow et al. (2016a). Finally, parameters are then updated according to the equation 1 where $0 < \alpha < 1$ is called the learning rate. Depending on the complexity of the dataset and the task, it might require hundreds of passes (called *epoch*) over the input dataset for convergence.

$$\mathcal{W}^{t+1} = \mathcal{W}^t - \alpha \nabla_{\mathcal{W}^t}^{\mathcal{L}^t} \tag{1}$$

## C  GRADIENT CLIPPING

Gradient Clipping (GC) is a method that is mainly known to help mitigate the problem of exploding gradients during training Goodfellow et al. (2016b). GC forces the gradients to fall within a narrow interval. There have been some efforts to analyze GC with respect to convergence. Authors in Zhang et al. (2019) prove (assuming a fixed step size) that training with GC can be arbitrarily faster than training without it. Moreover, the theoretical analysis suggested that small clipping values can damage the training performance. However, in practice that is rarely the case. Chen et al. (2020) has an interesting theoretical analysis coupled with empirical evidence (symmetry of gradients distribution w.r.t SGD trajectory) that answers the gap between previous theoretical and practical observations.

## D  INTEGRITY PROOFS

### D.1  RANDOM MINI-BATCH VERIFICATION

**Definition 2.** Define random variable $X = \sum_{b=1}^{\lceil B \times p_v \rceil} V_b$ to be the total number of random verification done. Here $V_b$ is 1 if the batch is chosen for verification but failed the verification (due to malicious computation). Please note that we need to catch at least one deviation with probability greater than $p_i$, and invalidate the overall model learning.

*Proof.* Proof of theorem 1

$$
\begin{aligned}
P(X \geq 1) = 1 - P(X = 0) &\geq p_i \\
1 - p_i &\geq P(X = 0) \\
1 - p_i &\geq \binom{\lceil B \times p_v \rceil}{0} p_c^0 (1 - p_c)^{\lceil B \times p_v \rceil} \\
1 - p_i &\geq (1 - p_c)^{\lceil B \times p_v \rceil} \\
\log(1 - p_i) &\geq \lceil B \times p_v \rceil \log(1 - p_c) \\
p_v &> B^{-1}\left(\frac{\log(1 - p_i)}{\log(1 - p_c)} - 1\right)
\end{aligned}
\tag{2}
$$

$\square$

As shown in Figure. 5 it is only required to verify a much smaller subset of the batch computations inside the TEE to ensure a high probability of correct computation. For example, for large datasets such as Imagenet Deng et al. (2009), it is needed to verify less than %1 of computation to have *0.9999* guarantee (when corruption probability is %0.5) on the correctness of computation outsourced to the GPU.

### D.2  RANDOM MINI-BATCH VERIFICATION WITH RANDOMIZED MATRIX MULTIPLICATION

**Definition 3.** Define random variable $V_b' = 1 \ if R_b = 1 \ \wedge \ MM\_verify(b) = 0 \ otherwise \ 0$ and $X = \sum_{b=1}^{\lceil B \times p_v \rceil} V_b'$. We need to detect at least one deviation with probability greater than $p_i$ while conducting random matrix multiplication verification.

Table 3: TEE Architectures Used

| Arch | FC1 | FC2 | FC3 |
|------|-----|-----|-----|
| VGG11 | (128,10) | (128,64,10) | (256,128,10) |
| VGG13 | (128,10) | (128,64,10) | (256,128,10) |
| VGG16 | (128,10) | (128,64,10) | (256,128,10) |

*Proof.* Proof of theorem 2

$$
\begin{aligned}
P(X \geq 1) &\geq p_i \\
1 - P(X = 0) &\geq p_i \\
1 - p_i &\geq \binom{\lceil B \times p_v \rceil}{0} (p_c(1-\alpha))^0 ((1-p_c) + p_c\alpha)^{\lceil B \times p_v \rceil} \\
1 - p_i &\geq ((1-p_c) + p_c\alpha)^{\lceil B \times p_v \rceil} \\
\log(1 - p_i) &\geq \lceil B \times p_v \rceil \log((1-p_c) + p_c\alpha) \\
p_v &> B^{-1}\left(\frac{\log(1-p_i)}{\log((1+(\alpha-1)p_c)} - 1\right)
\end{aligned}
\tag{3}
$$

□

## E    VERIFICATION PROBABILITY GROWTH WITH RESPECT TO DETECTION PROBABILITY

Fig. 5 shows how verification probability changes with respect to the probability that a batch step is maliciously manipulated by the attacker. First row shows the verification probability for a dataset with 60K samples. Second row depicts the required for much bigger dataset (1M samples) over different mini-batch sizes. The smaller the mini-batch size is, there is a higher chance for detecting malicious behavior.

## F    EXPERIMENTAL EVALUATION

### F.1    ENCLAVE HEAP SIZE IMPACT ON TEE PERFORMANCE

As shown in Figure 6, which depicts the impact of the heap size on the performance of DNNs for a single SGD step, it can be seen that increasing the heap size way beyond the available hardware limit (around 92MB) can have a negative impact on performance especially in the case of the VGG architecture. This result is mainly due to 1) driver level paging, which needs to evict enclave pages that require an extra level of encryption/decryption and 2) extra bookkeeping for the evicted pages.

### F.2    TEE PERFORMANCE ON CIFAR10

Figure 7 shows throughput performance for CIFAR10 dataset and 9 different VGG architectures (Table 3). We chose three VGG(11,13,16) architectures adapted for CIFAR10 image inputs with custom fully connected layers attached to its end. For CIFAR10, we generally do not benefit from randomized matrix multiplication scheme as well as ImageNet. Mainly it is because most of the operations and network layers fit well within the hardware memory limit. Therefore, since dimensions of MM operations are not too big, it does not improve significantly to use randomized matrix multiplications.

### F.3    IMPACT OF GRADIENT CLIPPING FOR HONEST TRAINERS

One important question is whether the gradient clipping used to prevent attacker to change parameters in a given mini-batch update would have performance impact during training session where there is no attack. Six experiment configurations were repeated 5 times each with different randomness (initialization, batch order, etc.). Initial learning rates are set $\in (0.1, 0.01)$ and clipping

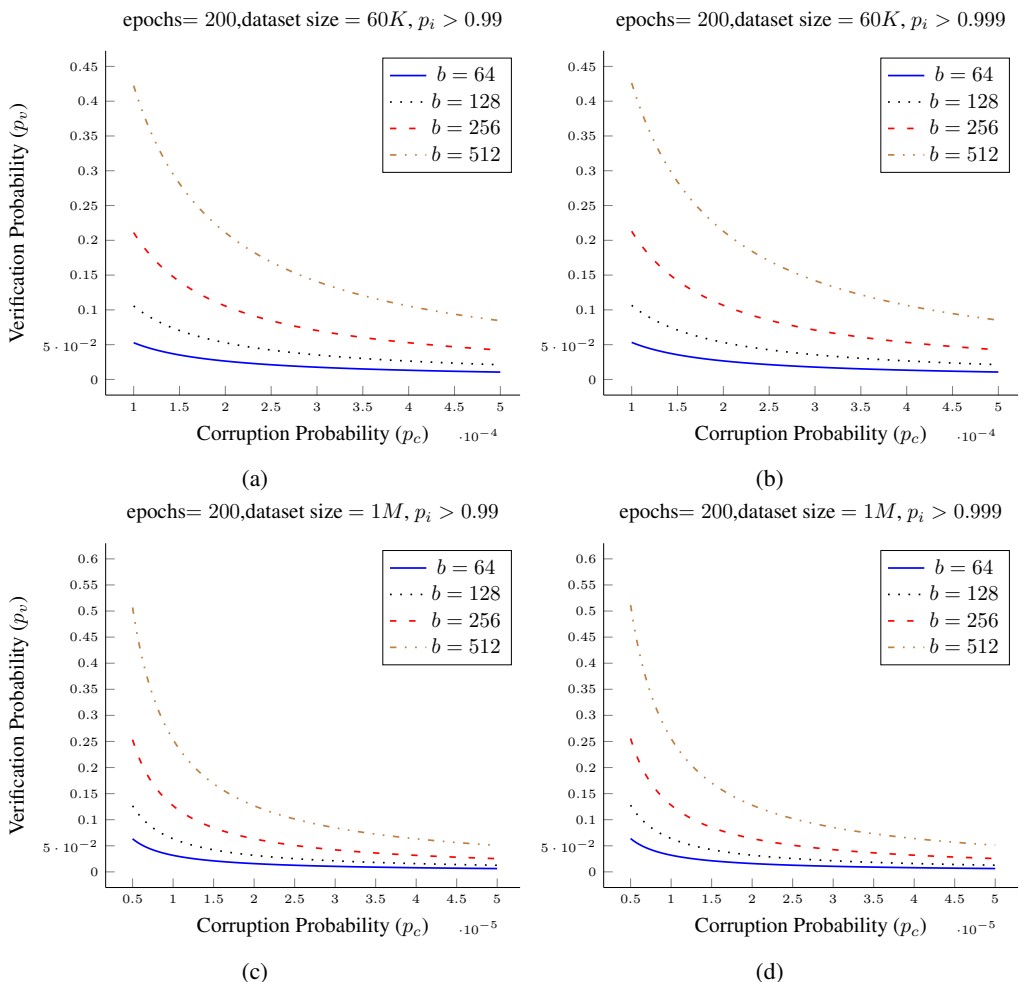

Figure 5: Required verification probability with respect to batch corruption probability and the desired integrity probability for a fixed 200 epochs and different SGD batch size.

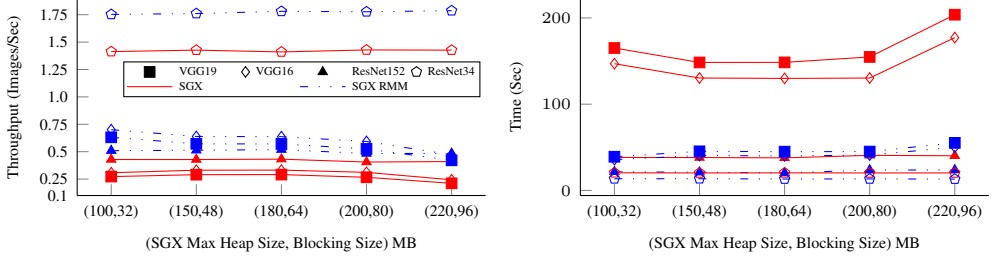

(a) Available heap with respect to throughput

(b) Available heap with respect to time spent on matrix-matrix(vector) multiplication

Figure 6: The impact of increasing TEE heap size on (a) overall throughput and (b) the time spent in matrix multiplication routine . VGG shows significant reduction in performance as opposed to ResNet.

thresholds are set $\in (nil, 0.001, 0.0005)$. In total, there are 30 ResNet56 (complex architecture with state-of-the-art performance) models trained on the CIFAR10 dataset (with no attack) for 200 epochs. It is shown that the clipping rate have very little impact on the test set accuracy, given an appropriate initial learning rate is chosen. Usually, for *SGD* Robbins & Monro (1951) optimizer

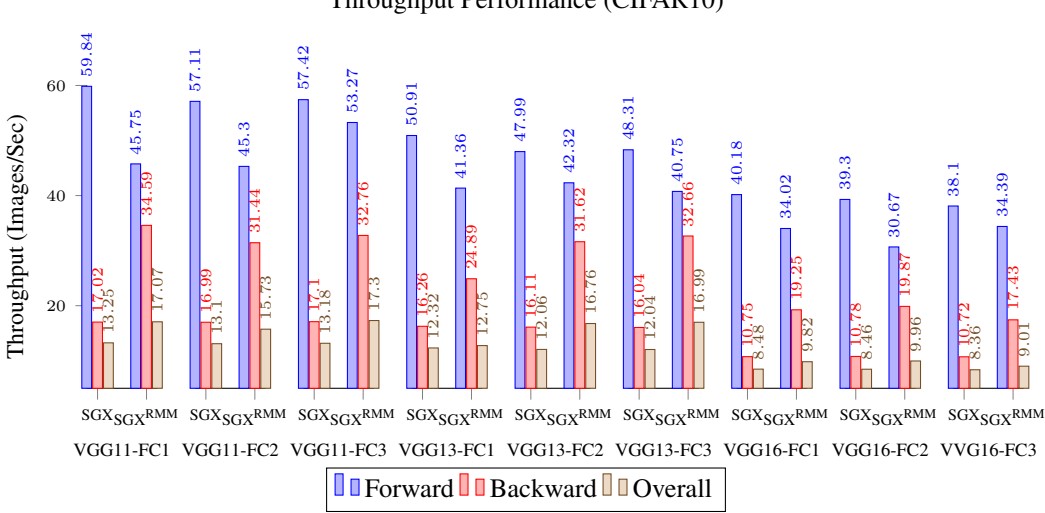

Figure 7: Throughput of SGD training step for VGG19,VGG16, ResNet152, and Resnet34 on CIFAR10 dataset. Randomized Matrix Multiplication can make verification twice faster in case of VGG architecture.

with momentum, a value of $0.1$ is chosen, (and for *Adam* optimizer a value less than $0.001$). For these experiments, we used the configuration with unbounded gradient updates as the main reference point. For learning decay schedule, we used a fixed decay by tenfold at epochs (50, 90, 130,160).

In figure 8a describes the mean and standard deviation (dashed lines) of test accuracy taken for 5 repetitions at each epoch for the two learning rate configurations. As it can be seen both models start to take giant leaps toward convergence at the first two learning decays enforced by the scheduler. Please note that these are reference runs that no gradient clipping is applied during the update step. Toward the end of training, the setting with the higher initial learning rate slightly performs better in terms of accuracy (the y-axis is not in % scale).

In figure 8b, for each composition of learning rate, and clipping value, the highest difference (accuracy rate) with respect to reference run is plotted. The plot shows test accuracy is not influenced so much by the clipping value, rather, it is highly dependent on the learning rate value. when $lr = 0.1$, both clipping values can achieve values that are close to the reference runs that has no gradient clipping, however, slightly smaller (most epochs it is negative, except jumps in the start). In figure 8c, the opposite of the previous measure is plotted. Again, by the end of the training, the gaps are significantly tightened for the case where a better learning is chosen. Therefore, having a smaller clipping value is not really impacting the performance in any considerable way.

Overall, figure 8, shows that clipping does not really impact the learning task negatively, once a good learning rate is chosen. One can observe that if the trainer choose an acceptable learning rate for the task, small clipping values such as $0.001$ or $0.0005$) does not impede the learning task. Once the model passes the first learning rate decay schedule, all the configuration behave the same in terms of their test performance compared to their reference model (no gradient clipping limit).

## F.4    ALL BACKDOOR TRIGGER EXAMPLES

## G    MATRIX MULTIPLICATION OPS OF COMMON DNN LAYERS

Table. 4 shows common MM operations in DNNs. Connected and convolutional layers use MM routines to compute feed forward output, parameter gradients, and input gradients.

Table 4: Matrix Multiplication Operations

| Layer Type | Pass | Computation | Verification | (Sub)Batched/ Precomp. |
|---|---|---|---|---|
| Fully Connected | Forward | $\mathcal{O}_{[B][O]} = \mathcal{I}_{[B][I]} \times (\mathcal{W}_{[O][I]})^\intercal$ | $\Upsilon_{[I][1]} = (\mathcal{W}_{[O][I]})^\intercal \times \mathcal{R}_{[O][1]}$ 
 $\mathcal{Z}_{[B][1]} = \mathcal{I}_{[B][I]} \times \Upsilon_{[I][1]}$ 
 $\mathcal{Z}'_{[B][1]} = \mathcal{O}_{[B][O]} \times \mathcal{R}_{[O][1]}$ | YES / YES |
| | Backward Parameters Gradient | $\nabla^{\mathcal{W}}_{[O][I]} = (\nabla^{\mathcal{O}}_{[B][O]})^\intercal \times \mathcal{I}_{[B][I]}$ | $\Upsilon_{[B][1]} = \mathcal{I}_{[B][I]} \times \Upsilon_{[I][1]}$ 
 $\mathcal{Z}_{[O][1]} = (\nabla^{\mathcal{O}}_{[B][O]})^\intercal \times \Upsilon_{[B][1]}$ 
 $\mathcal{Z}'_{[O][1]} = \nabla^{\mathcal{W}}_{[O][I]} \times \mathcal{R}_{[I][1]}$ | YES / NO |
| | Backward Inputs Gradient | $\nabla^{\mathcal{I}}_{[B][I]} = \nabla^{\mathcal{O}}_{[B][O]} \times \mathcal{W}_{[O][I]}$ | $\Upsilon_{[O][1]} = \mathcal{W}_{[O][I]} \times \mathcal{R}_{[I][1]}$ 
 $\mathcal{Z}_{[B][1]} = \nabla^{\mathcal{O}}_{[B][O]} \times \Upsilon_{[O][1]}$ 
 $\mathcal{Z}'_{[B][1]} = \nabla^{\mathcal{I}}_{[B][I]} \times \mathcal{R}_{[I][1]}$ | YES / YES |
| Convolutional | Forward | $\mathcal{O}_{[f][w_o.h_o]} = \mathcal{W}_{[f][k^2.C_i]} \times \mathcal{I}_{[k^2.C_i][w_o.h_o]}$ | $\Upsilon_{[1][k^2.C_i]} = \mathcal{R}_{[1][f]} \times \mathcal{W}_{[f][k^2.C_i]}$ 
 $\mathcal{Z}_{[1][w_o.h_o]} = \Upsilon_{[1][k^2.C_i]} \times \mathcal{I}_{[k^2.C_i][w_o.h_o]}$ 
 $\mathcal{Z}'_{[1][w_o.h_o]} = \mathcal{R}_{[1][f]} \times \mathcal{O}_{[f][w_o.h_o]}$ | NO / YES |
| | Backward Parameters Gradient | $\nabla^{\mathcal{W}}_{[f][k^2.C_i]} = \nabla^{\mathcal{O}}_{[f][w_o.h_o]} \times (\mathcal{I}_{[k^2.C_i][w_o.h_o]})^\intercal$ | $\Upsilon_{[w_o.h_o][1]} = (\mathcal{I}_{[k^2.C_i][w_o.h_o]})^\intercal \times \mathcal{R}_{[k^2.C_i][1]}$ 
 $\mathcal{Z}_{[f][1]} = \nabla^{\mathcal{O}}_{[f][w_o.h_o]} \times \Upsilon_{[w_o.h_o][1]}$ 
 $\mathcal{Z}'_{[f][1]} = \nabla^{\mathcal{W}}_{[f][k^2.C_i]} \times \mathcal{R}_{[k^2.C_i][1]}$ | NO / NO |
| | Backward Inputs Gradient | $\nabla^{\mathcal{I}}_{[k^2.C_i][w_o.h_o]} = (\mathcal{W}_{[f][k^2.C_i]})^\intercal \times \nabla^{\mathcal{O}}_{[f][w_o.h_o]}$ | $\Upsilon_{[1][f]} = \mathcal{R}_{[1][k^2.C_i]} \times (\mathcal{W}_{[f][k^2.C_i]})^\intercal$ 
 $\mathcal{Z}_{[1][w_o.h_o]} = \Upsilon_{[1][f]} \times \nabla^{\mathcal{O}}_{[f][w_o.h_o]}$ 
 $\mathcal{Z}'_{[1][w_o.h_o]} = \mathcal{R}_{[1][k^2.C_i]} \times \nabla^{\mathcal{I}}_{[k^2.C_i][w_o.h_o]}$ | NO / YES |

## H  GINN BLOCKING OF BIG MATRICES

By default, GINN allocates/releases resources on a per layer basis. In the event that even for a single sample, it is not possible to satisfy the memory requirements of network (either large network or large inputs), GINN breaks each layer even further.

For convolutional layers, the main memory bottleneck is *im2col*[3] which converts the layer's input (for each sample) of size $c_i \cdot w_i \cdot h_i$ to $[k^2 \cdot c_i] \times [w_o \cdot h_o]$ (k is kernel window size) matrix for a more efficient matrix multiplication. GINN divides the inputs across the channel dimension and processes the *im2col* on maximum possible channels that can be processed at once.

For fully-connected layers, the main memory bottleneck is the parameters matrix $\mathcal{W}_{[O] \cdot [I]}$ that does not depend on the batch size. GINN divides the matrix across the first dimension (rows), and processes the outputs on the maximum possible size of rows that fits inside the TEE for the corresponding layer.

---

[3]extracts redundant patches from the input image and lays in columnar format

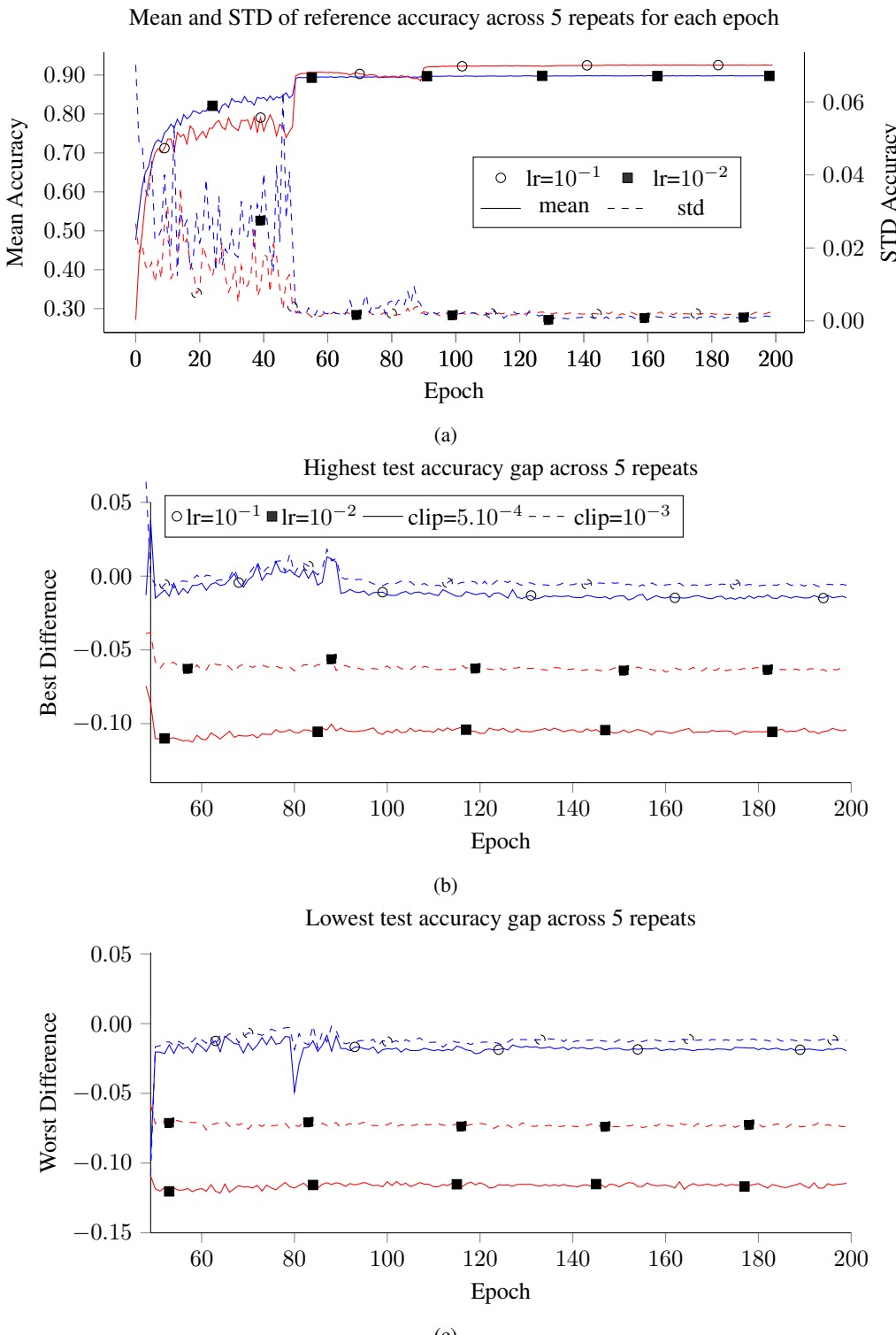

(a)

(b)

(c)

Figure 8: 8a Reference Models (no gradient clipping) mean/std on test accuracy of 5 repeats for two different learning rates. Each configuration had 5 repeats and a reference model (no attack and unbounded updates). 8b For each run configuration the test accuracy difference $diff_{lr,clip}$ is defined as $max\left(acc_{lr,clip}^{rep} - acc_{ref}^{rep}\right) \quad \forall rep \in [1,5]$. 8c $min\left(acc_{lr,clip}^{rep} - acc_{ref}^{rep}\right) \quad \forall rep \in [1,5]$

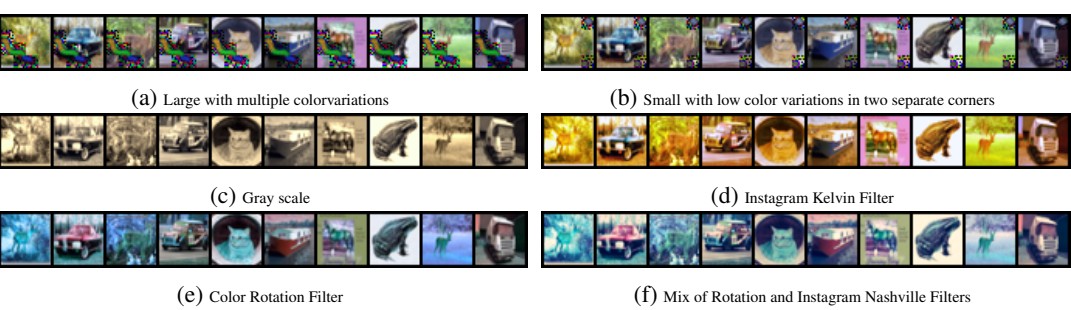

(a) Large with multiple colorvariations

(b) Small with low color variations in two separate corners

(c) Gray scale

(d) Instagram Kelvin Filter

(e) Color Rotation Filter

(f) Mix of Rotation and Instagram Nashville Filters

Figure 9: All examples of triggers on CIFAR10 images

