# OpenReview forum: "GINN: Fast GPU-TEE Based Integrity for Neural Network Training"
_ICLR.cc/2021/Conference — Reject_

### Official Review · AnonReviewer4 · 2020-10-26
**Straight Forward Integrity Checking**

**Rating:** 3
**Confidence:** 4

**Review:**

The paper discuss how to detect erroneous steps in gradient descent on a non-trusted GPU using a separate slower trusted execution environment,
by randomly deciding in each step whether to check the values returned by the GPU, as well as using small learning rates and clipping the gradients to ensure all updates are small.

The reason for checking this is based on the assumption that since GPU calculations are out sourced there may be trust issues and attackers with control of the
values returned by the GPU can alter the final network in subtle ways.

The paper includes experiments and shows that this approach is faster than just running everything on the trusted execution environment.
The experiments test an attack approach where the attacker tries to inject some bad samples to get some success of making the network being trained output some particular class on a particular image kind.


I have questions about the model. It is explained that the attacker is full control of CPU and GPU etc. but it is also assumed the attacker runs the code on GPU as expected. I would prefer a complete black box model, where the system can input model, batches, parameters etc. to the GPU and get whatever it wants back, i.e. gradients, activations, whatever you desire and given an input the attacker can decide to return whatever you want.

It seems the attacker is seemingly completely oblivious to the actual network, and parameters being used, and only measures his current success rate, and seems very naive.

Finally, the time is measure compared to a pure based TEE solution which is stated in the paper as completely unsatisfactory, and should instead be compared with training time without using any form of verification as this is the target.

In my opinion this paper is an implementation of a straight forward idea and the theorems for setting the probability parameters  are basic probably computations.
There is no theoretical contribution in the paper and all the arguments are heuristically based on some intuition about training dynamics.

In short, in my opinion the paper is simply not relevant or strong enough to warrant acceptance at ICLR.

---

> ### Author Response · Authors · 2020-11-25
> **Response to the points raised by the reviewer**
>
> I have questions about the model. It is explained that the attacker is full control of CPU and GPU etc. but it is also assumed the attacker runs the code on GPU as expected. I would prefer a complete black box model, where the system can input model, batches, parameters etc. to the GPU and get whatever it wants back, i.e. gradients, activations, whatever you desire and given an input the attacker can decide to return whatever you want.
>
> Response:
> Model is initialized within the SGX. Every time an SGD step is performed, the gradients are returned to the SGX for clipping, and updates are applied within the SGX (TEE we used in our experiments). Model’s hash is always authenticated whenever an update happens, and a new authentication code is generated for the new model weights. The only way the attacker can influence the model is through sending wrong updates. To prevent that we randomly check (Poll) the SGD step within the SGX. For that, we only need the input batch, parameters values (from last SGD update) and current reported gradients to see if the reported values match the computed ones inside the SGX. However, it is theoretically possible to check the computation at a finer grain such as a specific activation neuron, parameters within a kernel in some layers, etc. In fact, it might give a better theoretical bound since the more computational steps (dividing the SGD steps into smaller sub-steps) for a fixed number of malicious steps might mean the less verification! However, our proposed approach allows a much easier implementation and analysis.
>
> It seems the attacker is seemingly completely oblivious to the actual network, and parameters being used, and only measures his current success rate, and seems very naive
>
> Response:
>
> Please note that our defense mechanism is strategy proof since we randomly check we do not care how and when the attacker attacks, we only care that such an attack step occurs not so rarely.  In other words, the simple random verification strategy advocated by our work does not need to run experiments with sophisticated attackers, hence our defense mechanism will not be impacted at all by the different attacks that require a similar number of modification/attack steps. In our theoretical analysis, we show the detection rate based on the number of attack steps and the verification rate.
>
> We would like to emphasize that we are the first ones to evaluate the simple random verification approach combined with gradient clipping to substantially improve the training time while preserving integrity with high probability, and provide an important baseline against any type of attack and future work.
>
> Finally, the time is measure compared to a pure based TEE solution which is stated in the paper as completely unsatisfactory, and should instead be compared with training time without using any form of verification as this is the target.
>
> Response:
>
> Without any verification, the GPU based training would be 50-200X times faster than pure TEE based solution. Please note that our goal is to improve the state of art in integrity preserving training in cloud settings. Hence, we compare our solution to a pure TEE based solution that provides integrity.

---

### Official Review · AnonReviewer2 · 2020-10-27
**Official Blind Review #2**

**Rating:** 5
**Confidence:** 3

**Review:**

It seems that the paper is focusing on the privacy preserving training of deep neural networks by developing a Learning-as-a-Service framework. It assumes that all resources may be penetrated by adversaries except the TEE. In this paper, the authors leverage random verification to detect the attacks and shows how gradient clipping can defend against attacks.

As an avid user of cloud training, I can agree upon the motivation of the paper that the integrity of training models in the cloud. However, I want to write about some issues that I have about the paper.

First, it seems that the main contribution of the paper is two folds (1) using randomized verification over full verification and (2) verifying gradient clipping to defend against the attacks. However, mere randomized runs of verifications speak for a rather humble contribution. It would be interesting to see whether there can be other more intelligent methods that can identify them. It would add to the novelty and merit if there were some evaluations too for this.

Furthermore, to verify the real effectiveness of the gradient clipping as the defense mechanism, it would be nice to see some theoretical analysis on this front.

Furthermore, I have some issues with the evaluation. First, the paper compares to a full verification in Figure 2. However, this information is given without any breakdown of "actual training" vs. "verification". This makes it hard for the readers to see exactly where the speedups come from and where.

The paper does not seem to demonstrate any evaluation n terms of the potential accuracy loss for large datasets. While F.3 seems to demonstrate this on CIFAR10, this cannot fully verify that such gradient clipping wont hurt the accuracy of the models.

I am open to reevaluating the paper given the authors' feedback.

============================================

I thank the authors' for their response.
I am satisfied with the answer regarding gradient clipping. As such, I raised my score to 5.

However, I still cannot get away from the thought that random verification seems to speak for a rather humble a contribution. Combined with the lack of breakdown of "actual training" vs. "verification", I am rather hesitant to give a score higher than this. Regarding the experimentation on ImageNets, I understand that there was lack of time to add more experiments on this front. However, it would provide a more concrete message if the paper includes these additional experiments.

---

> ### Author Response · Authors · 2020-11-25
> **Response to Extra Experimentation**
>
> However, mere randomized runs of verifications speak for a rather humble contribution. It would be interesting to see whether there can be other more intelligent methods that can identify them. It would add to the novelty and merit if there were some evaluations too for this.
>
> Response:
>
> In our experiments, we logged the norm of  gradients (before clipping) and how it changes when the attack is starting. If we use these signals to devise a defense, the attacker can also adapt its attack to hide these signals. Therefore, we believe that random verification is robust to any type of attack. Our only requirement, as we showed using probabilistic analysis, is that the attacker needs to attack at multiple steps. Hence, simple random verification advocated by our work is strategy proof against different attacks.  Therefore, this simple strategy is more robust as long as the attack requires multiple steps.
>
>
> Furthermore, to verify the real effectiveness of the gradient clipping as the defense mechanism, it would be nice to see some theoretical analysis on this front.
>
> Response:
>
> As we discuss in our response to reviewer 1, no clipping clearly makes the attacker’s job easier. It can even match the desired model parameters that allow successful attack in a single step.  With new experiments added, we show empirically that low clipping rates would require more steps. Please see the updated experimental section.
>
> Intuitive reasoning for low clipping rate is that imagine the attacker wants some model parameters to be $\theta’$ for successful attack. Given the current model parameters $\theta$, it can only move towards $\theta’$ bounded by the clipping value and learning rate. Hence, at each step, it can move the model parameters less with a lower clipping rate.
>
> The paper does not seem to demonstrate any evaluation in terms of the potential accuracy loss for large datasets. While F.3 seems to demonstrate this on CIFAR10, this cannot fully verify that such gradient clipping won't hurt the accuracy of the models.
>
> Response:
>
> We agree that the accuracy may be lower in some certain cases. In our experiments, we used a fixed learning rate schedule and on CIFAR and other datasets we tried we did not observe any loss.  Our framework suggests possible solutions to this issue. For example, higher clipping rates could be combined with higher verification inside TEE steps.

---

### Official Review · AnonReviewer1 · 2020-10-29
**The paper targets security challenges of deep neural networks. The authors provide evaluation results to demonstrate their work but more details are needed in some discussions.**

**Rating:** 6
**Confidence:** 3

**Review:**

The paper targets security challenges of deep neural networks. While solutions can hardly scale up to support realistic DNN model training workloads, the authors propose GINN to support integrity-preserving DL training by random verification of stochastic gradient steps inside trusted execution environments (TEE). GINN combines random verification and gradient clipping to achieve good performance. The experimental results show that GINN achieves 2x-20x performance improvement compared with the pure TEE based solution while guaranteeing the integrity with high probability.

#######################
Pros
1.	The security of deep neural networks is a very interesting and challenging topic.
2.	The paper is well structured, and the assumptions are clear.
3.	The authors provide extensive evaluation results to demonstrate their work.

#######################
Cons

1.	The authors illustrated how to do probabilistic verification: randomly decide to verify the computation over each batch. But how is a batch verified inside the TEE? It is not clearly stated in the paper.
2.	When discussing the limitation of existing solutions, the authors claim that they are evaluated with small datasets (MNIST, CIFAR10). But there is not much evaluation on large datasets for GINN either. Maybe adding an experiment on a large dataset in Figure 3 will be helpful?
3.	The random verification strategy is based on the observation that it is unnecessary to verify all of the computation steps. Where does the observation come from? More details, data or citations will help.

#######################
Minor comment
1.	As far as I understand, the use of gradient clipping is intended to reduce the bias introduced by the adversary (which might be missed by random verification ) so that the adversary tends to launch multiple attacks? It would be better to explain that clearly in the paper.
2.	It would be good to see some results about Verification Rate v.s. clipping rate.

---

> ### Author Response · Authors · 2020-11-25
> **Response to various questions**
>
> The authors illustrated how to do probabilistic verification: randomly decide to verify the computation over each batch. But how is a batch verified inside the TEE? It is not clearly stated in the paper
> Response:
> Same algorithms (layers, forward and backward steps) that run on the GPU are repeated within the enclave. Please note that SGX has the seed to derive the input batch, plus the randomness for layers needed. If the computed gradients in the verification round do not match the reported ones, it identifies a misstep. We also have updated the figure caption and main text to reflect this point.
>
> Large dataset experimentation will be amended
>
> Response:
>
> We added experiments with ImageNet as well. Also, please note that irrespective of the dataset, the random verification strategy improves the overall run time since only some subset of the steps would be verified inside the slow TEE.
>
> The random verification strategy is based on the observation that it is unnecessary to verify all the computation steps. Where does the observation come from? More details, data, or citations will help.
>
> Response:
>
> This is the observation of our work. Basically, gradient clipping is important to prevent the attacker from attacking in a few steps. For example, if the target model parameter is $\theta’$ and the regular training learns model parameters $\theta$, then it is clear that without clipping the attacker can easily move $\theta$ to $\theta’$ in one step. Hence, clipping is needed.  Please see the updated experiment section for more experiments on the impact of clipping.
>
> As far as I understand, the use of gradient clipping is intended to reduce the bias introduced by the adversary (which might be missed by random verification ) so that the adversary tends to launch multiple attacks? It would be better to explain that clearly in the paper.
>
> Response:
>
> We mention this aspect in the introduction.
>
> It would be good to see some results about Verification Rate v.s. clipping rate.
>
> Response:
>
> We chose very small values for clipping rate for our analysis. Initial experiments with larger clipping rates made the attack easier, hence it required much higher verification rates. So, we only focused on training and experimenting tight clipping intervals for the majority of experiments. However, for the case of ImageNet, we chose two values: high( 2.0) and low (0.0005) which are significantly different in magnitude and investigate their potential impact under different attack scenarios. Our results reported in the update experimental section indicate that low clipping rate requires more attack steps by the attacker.

---

### Official Review · AnonReviewer3 · 2020-10-30
**Practical approach for secure training using a TEE**

**Rating:** 7
**Confidence:** 4

**Review:**

Paper Summary:

The paper presents new techniques to enable secure neural training in the TEE+GPU paradigm. This is a natural extension of previous work like Slalom which only handles the inference case. The authors propose a two-step approach. First, they clip the gradients during training to force the attacker to insert multiple deviations to influence the model. They then randomly verify the integrity a subset of the gradient updates to check for tampering. Combining these two ideas the authors demonstrate a 2-20x improvement over a TEE only benchmark.

Score Rationale:

- The paper presents two novel ideas
  - The use of probabilistic checks on a random subset of the gradient update steps
  - The use of gradient clipping to force the attacker to tamper with multiple update steps to affect the final model
- While the core idea of verifying the linear layers is a direct extension of previous work, the probabilistic checks offer a significant improvement in performance

Strengths:

- The paper presents a detailed empirical evaluation of the required verification rate as function of multiple hyper-parameters such as the style of the injected error, aggressiveness, and timing of the poisoning
- The paper demonstrates that gradient clipping has minimal impact on the network accuracy if a suitable learning rate is chosen

Weaknesses:

- The core contribution of this work relies on the effectiveness of clipping procedure in forcing the attacker to introduce a large number of faults for a successful attack. However, this conclusion is not directly supported by empirical evidence, i.e. there is no ablation study of the verification rate required in the absence of clipping.

Minor Comments/Questions:

- There is a typo in the caption of Fig. 6 which should specify the CIFAR-10 dataset rather than ImageNet

---

> ### Author Response · Authors · 2020-11-25
> **Regarding ablation analysis**
>
> Gradient clipping is important to prevent the attacker from attacking in a few steps. For example, if the target model parameter is $\theta’$ and the regular training learns model parameters $\theta$, then it is clear that without clipping, the attacker can easily move $\theta$ to $\theta’$ in one step. Given the plethora of trojan attacks, it would be difficult to come up with a mechanism to detect faulty gradient updates in case the attacker decides to send huge updates and modify the model in O(1) attempts.
>
> To show the importance of gradient clipping, we conducted various experiments on ImageNet. We show the impact of gradient clipping on the case where there is no attack and where there is attack. Please see the updated experimental section for the ablation study. The results indicate that clipping is important for convergence in a no-attack setting, and requires the attacker to attack in more steps while minimal impact on accuracy.
>
> We fixed all the typos pointed out.

---

### Decision · Program_Chairs · 2021-01-07
**Final Decision**

**Decision:**

Reject

**Comment:**

While all reviewers agree the problem of TEEs for model training is well motivated, the reviewers remain divided on whether the concept of randomly selecting computations to verify has sufficient novelty, and whether the proposed gradient clipping method is well motivated.